# A Theoretical Study of the Jacobian Matrix in Deep Neural Networks

## Abstract

Due to the compositional nature of neural networks, increasing their depth can lead to issues of vanishing or exploding gradients if the initialization scheme is not carefully selected (Poole et al., 2016; Schoenholz et al., 2017; Hayou et al., 2019). One approach to identifying a desirable initialization scheme involves analyzing the behavior of the input-output Jacobian and ensuring that it does not degenerate exponentially with depth. Such an analysis has been conducted in previous works, such as Pennington et al. (2017), where the authors discovered a critical initialization scheme that ensures Jacobian stability, as confirmed by empirical results. The analysis carried in such studies is limited to initialization and leverages classical results in random matrix theory. In this paper, we extend this analysis beyond initialization, and study Jacobian behaviour during training. Notably, we show that a notion of stability holds throughout training (if satisfied at initialization), hence providing a theoretical explanation for the crucial role of initialization. To do this, we first prove a general theorem that utilizes recent breakthrough results in random matrix theory (Brailovskaya and van Handel, 2022). To show the broad applicability of our framework, we also provide an analysis of the Jacobian in other scenarios such as sparse Networks and non-iid initialization.

## 1 Introduction

Deep neural networks (DNNs) have revolutionized machine learning, achieving state-of-the-art results in many applications (LeCun et al., 2015). However, despite their impressive performance, training DNNs remains a challenging task and often requires heavy hyper-parameters tuning. One important factor that affects the trainability of DNNs is the behaviour of the input-output Jacobian, which measures the sensitivity of the network's output to changes in its input(Saxe et al., 2013). Improper initialization can lead to issues of vanishing or exploding Jacobians/gradients (as depth increases), which might cause the network to get stuck in poor local optima or diverge during training.

Prior research on the Jacobian in DNNs has concentrated on its behaviour at initialization (Pennington et al., 2017; Collins and Hayase, 2023), where the network weights are initialized, e.g. with independent and identically distributed (i.i.d) Gaussian weights, or orthogonal weights (e.g., Pennington et al. (2017); Hanin and Nica (2020); Chhaibi et al. (2022)), and showed both theoretically and empirically a correlation between 'stable' Jacobian behaviour at initialization and favorable performance properties (training and generalization). Similar works have focused on studying other related quantities such as covariance kernel and gradients at initialization to derive the 'Edge of Chaos' initialization scheme (Poole et al., 2016; Schoenholz et al., 2017; Pennington et al., 2017; Hayou et al., 2019), which guarantees gradient stability and better information propagation. However, the analysis in these works is limited to initialization and fails to cover other scenarios, e.g. what happens to the Jacobian during training? or what happens to the Jacobian after we prune the network (LeCun et al., 1990; Hassibi et al., 1993; Lee et al., 2018b; Hayou et al., 2021; Wang et al., 2020; Frankle and Carbin, 2019a)? etc.

In this paper, we propose to (partially) address some of these questions by developing a general theoretical framework (stability theorem, Appendix A) based on recent breakthroughs in random matrix theory, on the products of random matrices in non-iid settings (Brailovskaya and van Handel, 2022). Our primary objective is to *gain insights* into the behaviour of the Jacobian under different

scenarios in the simple case of Multi-Layer Perceptrons (MLPs). To the best of our knowledge, this is the first theoretical work on the behaviour of Jacobian in DNNs beyond initialization. Extending these results to other architectures is challenging and is an interesting topic for future research.

Our analysis reveals several interesting phenomena. Firstly, with i.i.d initialized networks, if the Jacobian is 'stable' (we formally define stability later in the paper) at initialization, then, under some assumptions it remains roughly stable during training. Secondly, we show a three-phases behaviour of the Jacobian norm during training: a *stagnation* phase where the Jacobian norm remains roughly constant, followed by a *descent* phase where the Jacobian exhibits a rapid drop in norm, and a *convergence phase* where the Jacobian norm converges and remains roughly constant until convergence of gradient descent. Lastly, to show the broad applicability of our theoretical framework, we show that in sparse networks, the Jacobian behaves similarly to that of a full network (non sparse) if we scale the non-zero weights suitably, where the scaling factor depends on the pruning method. In Appendix F, we further provide an application of our framework in the case of networks with non-iid initialization weights, and find that there exists a width-dependent correlation threshold that guarantees similar Jacobian properties to the iid case. All the proofs are deferred to the Appendix.

## 2 SETUP AND DEFINITIONS

Throughout this paper, we consider the Multi-Layer Perceptron (MLP) architecture given by:

$$
\text{MLP} \triangleright \begin{cases} Y_0(x) = W_{in}x, \\ Y_k(x) = W_k\phi(Y_{k-1}(x)), \quad k = 1, \ldots, L, \\ Y_{out}(x) = W_{out}\phi(Y_L(x)), \end{cases} \tag{1}
$$

where $x \in \mathbb{R}^d$ is the input, $Y_{out} \in \mathbb{R}^o$ is the network output, $W_{in} \in \mathbb{R}^{n \times d}, W_k \in \mathbb{R}^{n \times n}, W_{out} \in \mathbb{R}^{o \times n}$ are the network weights, and $\phi$ is the ReLU activation function given by $\phi(z) = \max(z, 0)$ (acting coordinate-wise). For the sake of simplification, we omit here the bias terms in the definition of the MLP. We refer to $Y_k$ as the pre-activations or the features and $\phi(Y_k)$ as the activations. Hereafter, *width* and *depth* will be used to refer to $n$ and $L$, respectively.

In practice, neural networks are usually trained with gradient-based algorithms such as Stochastic Gradient Descent (SGD), Adam, etc. (LeCun et al., 2015; Kingma and Ba, 2014; Bottou, 2012). This requires the calculation of the gradients of some loss function with respect to the weights $(W_k)_{1 \leq k \leq L}$ using back-propagation. Let $\ell$ be a loss function (e.g. mean-squared error for regression, and cross-entropy loss for classification) and $\mathcal{D} = \{(x_i, z_i), i = 1 \ldots N\}$ be a fixed training dataset. DNN training aims to minimize the empirical objective $\mathcal{L}(\mathbf{W}) = N^{-1}\sum_{i=1}^N \ell(Y_{out}(x_i), z_i)$, where $\mathbf{W} = \{W_{in}, (W_k)_{1 \leq k \leq L}, W_{out}\}$. With GD, the parameters $\mathbf{W}$ are updated with the rule:

$$
\mathbf{W} \leftarrow \mathbf{W} - \eta\frac{\partial \mathcal{L}}{\partial \mathbf{W}}.
$$

For a datapoint $(x, z)$, the gradient of the loss function evaluated at $(x, z)$ w.r.t. the weights $W_k^{i,j}$ (for some $i, j \in \{1, \ldots, n\}$) is given by:

$$
\frac{\partial \ell(Y_{out}(x), z)}{\partial W_k^{i,j}} = \frac{\partial \ell(Y_{out}(x), z)}{\partial Y_k^i(x)}\phi(Y_{k-1}^j(x)) = \frac{\partial \ell(Y_{out}(x), z)}{\partial Y_L(x)}^\top \frac{\partial Y_L(x)}{\partial Y_k^i(x)}\phi(Y_{k-1}^j(x)).
$$

Hence, the gradients inherently depend on the Jacobian terms

$$
J_k(x) = \frac{\partial Y_L(x)}{\partial Y_{k-1}(x)} = \left(\frac{\partial Y_L^i(x)}{\partial Y_{k-1}^j(x)}\right)_{1 \leq i,j \leq n} \in \mathbb{R}^{n \times n},
$$

for $k \in \{1, 2, \ldots, L\}$. Using the chain rule, it is easy to see that for $k \in \{1, \ldots, L-1\}$, the Jacobian satisfies the recursion $J_k(x) = J_{k+1}(x) \times W_k D_{k-1}(x)$, where $D_k(x) = \text{Diag}(\phi'(Y_k(x))) \in \mathbb{R}^{n \times n}$. Thus, we can express the Jacobian terms as a product:

$$
J_k(x) = \prod_{l=k}^L W_l D_{l-1}(x), \quad k \in \{1, \ldots, L-1\}.
$$

Hereafter, we will denote the Jacobian without reference to its input, and use $J_1 = \prod_{l=1}^{L} W_l D_{l-1}$, as defined earlier. We assume a fixed non-zero input $x$ for the analysis. However, our empirical results will show that our findings hold for randomly selected inputs from the dataset, demonstrating that our conclusions are independent of the input choice[1].

Due to the nature of the Jacobian (product of matrices), one can anticipate the occurrence of vanishing or exploding gradient phenomena in instances where the weights are improperly initialized. By examining the spectral norm of the Jacobian, denoted by $\|J_1\|$ (i.e., the largest singular value of $J_1$), in relation to the depth $L$, distinct regimes can be identified wherein the Jacobian norm exhibits either exponential exploding or vanishing, or alternatively demonstrates a sub-exponential dependence relative to depth Pennington et al. (2017). It is the exponential dependence on the depth that poses a practical problem, as it typically leads to swift degradation of the gradients (exponential vanishing) and numerical instability (exponential exploding). When the depth dependence is sub-exponential, the network is deemed to be *stable*. Hereafter, we will use the notation $b_L = \Theta(a_L), b_L = \mathcal{O}(a_L), b_L = o(a_L)$, to refer respectively to $\alpha a_L \leq b_L \leq \beta a_L$ ($\alpha, \beta > 0$ constants), $|b_L| \leq \beta |a_L|$ for all $L$, and $\lim_{L \to \infty} b_L (a_L)^{-1} = 0$. We also use the notation $\tilde{\Theta}$ and $\tilde{\mathcal{O}}$ to hide any sub-exponential terms.

**Definition 1** (Stable Jacobian). *We say that the Jacobian of a network with a distribution $q$ over the weights[2] $\mathbf{W} \sim q$ is stable if:*

$$\mathbb{E}_{\mathbf{W} \sim q}[\|J_1\|] = \tilde{\Theta}_L(1),$$

*where $\|.\|$ denotes the spectral norm $\|A\| = \sqrt{\lambda_{\max}(AA^\top)}$.*

Here, we define stability for any weight distribution $q$, including the weight distribution during training (Section 4). Note also that here stability is defined as $\tilde{\Theta}(1)$ instead of $\Theta(1)$, which hides sub-exponential terms. This is because empirical results suggest that for typical network depths (e.g., in the range of 10 to 100), sub-exponential dependence does not significantly affect the performance (Poole et al., 2016; Schoenholz et al., 2017; Hayou et al., 2019). In the following, we will analyze the infinite-width limit of the Jacobian. In this limit, under some assumptions, the norm $\|J_1\|$ converges to a deterministic value almost surely (see Pennington et al. (2017) for the result at initialization, and Appendix A for a more general setup). As a result, taking the average over $\mathbf{W}$ has no effect in this limit and one can think of the stability condition as being similar to $\|J_1\| \approx \tilde{\Theta}_L(1)$.

## 3 JACOBIAN WITH I.I.D WEIGHTS AT INITIALIZATION

The large depth behaviour of the Jacobian has so far been studied at initialization, i.e., when the weights $\mathbf{W}$ are i.i.d randomly sampled from a fixed underlying distribution $q_0$. In particular, Pennington et al. (2017) showed that when wide neural networks are initialized with independent Gaussian weights $\mathcal{N}(0, \zeta)$, the largest singular value of the input-output Jacobian $J_1$ can either exponentially explode or vanish with depth if the variance of the weights is different from $\zeta = 2/n$, which is also known as *the Edge of Chaos* initialization. This choice of $\zeta$ guarantees stability at initialization. This result holds under the following approximation that simplifies the analysis.

**Approximation 1.** *In the infinite-width limit, the diagonal entries of $(D_k)_{k \in 0,...,L}$ behave as i.i.d Bernoulli variables with parameter $1/2$. Moreover, they are independent of the weights $\mathbf{W}$.*

It is easy to see why Approximation 1 is valid in the large-width regime. When $n \to \infty$, it is well known that the entries of the pre-activations $(Y_k^i)_{1 \leq i \leq n}$ converge (in distribution) to i.i.d Gaussian random variables that are independent across $i$ and $k$ (Neal, 1995; Lee et al., 2018a; Matthews et al., 2018; Hayou et al., 2019; Yang, 2019). Moreover, these entries become independent of the weights $\mathbf{W}$ in this limit. Hence, since the matrix $D_k$ consists of diagonal elements of the form $\phi'(Y_k^i) = 1_{Y_k^i > 0}$, it holds that the diagonal elements become approximately i.i.d Bernoulli random variables with parameter $1/2$ when $n$ is large. We refer the reader to Appendix G for an empirical verification of Approximation 1.

---

[1]It should be noted that while the input can impact stability in a DNN, this impact is often minor if the dataset is normalized, with the architecture and weight distribution playing a more significant role.

[2]Since neural networks are trained starting from a random initialization, then at any training stage, the weights $\mathbf{W}$ are random.

**Notation.** Hereafter, we use the approximate symbol "$\approx$" instead of "$=$" for any result derived under some approximation. Similarly, we use "$\lesssim$" instead of "$<$".

Now assume that the weights are initialized as $W_k^{ij} \sim \mathcal{N}(0, \sigma_w^2/n)$ for some $\sigma_w > 0$. Then, the following holds.

**Theorem 1** (Corollary of Eq. (17) in Pennington et al. (2017)).
*In the limit $n \to \infty$, under Approximation 1, we have the following:*

$$\|J_1\| \approx \Theta_L\left(L\left(\frac{\sigma_w^2}{2}\right)^L\right).$$

*In particular, the choice $\sigma_w^2 = 2$ guarantees stability.*

Note that the choice $\sigma_w^2 = 2$ corresponds also to the Edge of Chaos initialization; an initialization scheme that allows deeper signal propagation in MLPs Poole et al. (2016); Schoenholz et al. (2017); Hayou et al. (2019). Fig. 1 shows the Jacobian norm for different choices of $\sigma_w$ and depths $L$. Exponential exploding/vanishing with depth can be observed in the non-critical initialization cases $\sigma_w^2 \in \{0.5, 4.0\}$, while a sub-exponential growth w.r.t. depth is achieved with the critical initialization $\sigma_w^2 = 2$ as predicted by Theorem 1. In Appendix G, we report the accuracy of trained networks for varying architectures (MLP, VGG) and initialization schemes (resulting in different Jacobian norms) and further confirm that Jacobian stability is necessary to achieve non-trivial performance. This adds to the empirical evidence provided in Pennington et al. (2017); Chhaibi et al. (2022).

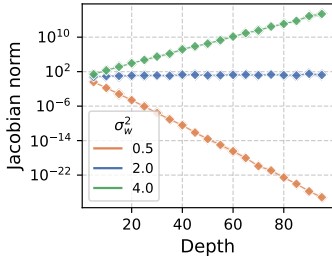

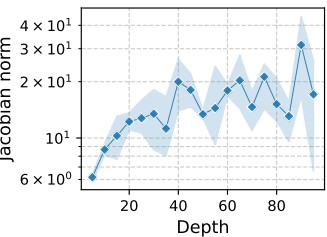

Figure 1: Illustration of the Jacobian norm at initialization in an MLP network of width $n = 256$ and varying depth. The input is randomly selected from MNIST. All results are averaged over 3 runs. **(Top)** Impact of depth on the Jacobian norm for different $\sigma_w$. **(Bottom)** Evolution of the Jacobian norm as a function of depth for critical initialization.

## 4 STABILITY DURING TRAINING

Empirical evidence suggests that critically initialized DNNs exhibit a significant performance boost (Poole et al., 2016; Schoenholz et al., 2017; Hayou et al., 2019). This observation suggests that the Jacobian remains stable (no exponential exploding/vanishing with depth) throughout the training process. Despite this, to the best of our knowledge, there is currently no theoretical explanation in the literature for this phenomenon. Using a variant of Approximation 1, a recent result from Random Matrix Theory (Brailovskaya and van Handel, 2022), and a concentration result from the Tensor Programs framework (Yang and Hu, 2021), we show that:

- If the Jacobian is stable at initialization, it remains stable throughout training.

- There exist three phases for Jacobian behaviour: the *stagnation phase* which is an initial training stage where (under some conditions on the learning rate) the Jacobian norm remains asymptotically constant. This initial phase appears when the learning rate is sub-optimal (see below for more details); the *decent phase*, a second training stage where the Jacobian norm decreases significantly without exhibiting any exponential exploding/decay. Using a toy model, we provide a heuristic explanation to this phenomenon; the *convergence phase*, a later training stage during which the Jacobian norm converges and remains roughly constant.

Hereafter, we use the superscript $t$ to denote the value of some object at training step $t$. Consider the full-batch gradient descent rule[3] given by:

$$W_l^{t+1} = W_l^t - \gamma \frac{\partial \mathcal{L}}{\partial W_l^t}, \quad l \in \{1, \ldots, L\}, \tag{2}$$

where $t \in \{0, \ldots, T\}$ is the training step, $\frac{\partial \mathcal{L}}{\partial W_l} \in \mathbb{R}^{n \times n}$ is the matrix with entries $\frac{\partial \mathcal{L}}{\partial W_l^{ij}}$, and $\gamma > 0$ is the learning rate. For the sake of simplification, we assume that the loss is given by

---

[3]To simplify the analysis, we consider the full-batch gradient descent instead of SGD.

$\mathcal{L}(\mathbf{W}) = (2n)^{-1}\|Y_L(x) - z\|^2$ for some (input, target) pair $(x, z)$.[4] All Jacobian matrices in this section are computed for the input $x$.

## 4.1 JACOBIAN STABILITY DURING TRAINING

Consider the Jacobian at time $t$, given by $J_1^t = \prod_{l=2}^L W_l^t D_{l-1}^t$. After one gradient step, we have

$$J_1^{t+1} = \prod_{l=1}^L \left( W_l^t - \gamma \frac{\partial \mathcal{L}}{\partial W_l^t} \right) D_{l-1}^{t+1} = \tilde{J}_1^t + \sum_{k=1}^L (-\gamma)^k \Gamma_k^t,$$

where $\tilde{J}_1^t = \prod_{l=2}^L W_l^t D_{l-1}^{t+1}$ and, for instance, $\Gamma_L^t = \prod_{l=1}^L \frac{\partial \mathcal{L}}{\partial W_l^t}$.

Note that $\tilde{J}_1^t$ is a *variant* of $J_1^t$, computed with matrices $D_l^{t+1}$ instead of $D_l^t$. Therefore, it becomes clear that a variant of Approximation 1 is needed to deal with the infinite-width limit.

**Approximation 2.** *Given some fixed $t$, in the infinite-width limit, the diagonal entries of $(D_k^{t'})_{k \in 0,\ldots,L}$ behave as i.i.d Bernoulli variables with parameter $1/2$ for all $t' \leq t$. Moreover, they are independent from the weights $\mathbf{W}^{t'}$. The weight matrices $\mathbf{W}_l^{t'}$ are approximately i.i.d across $l$.[5]*

Approximation 2 generalizes Approximation 1 in the context of training. It can be weakened by considering "free-probability" properties instead of independence, but we believe this is an unnecessary complication for this work. The approximation that $\mathbf{W}_l^{t'}$ are i.i.d across $l$ for some $t' \leq t$ is justified by the fact that the learning rate usually scales as $\mathcal{O}(n^{-1})$ (this is required in order to avoid any feature blow-up in the infinite-width limit) which becomes small in the infinite-width limit. We invite the reader to check Appendix G for a discussion and verification of this approximation.

**Theorem 2** (Jacobian during training). *Assume that the learning rate satisfies $\gamma = \mathcal{O}_{n,L}(n^{-1}L^{-3})$. Then, for any $t$, under Approximation 2 holding up to time $t$, we have that*

$$\lim_{n \to \infty} \|J_1^t\| - \|J_1^{iid}\| \approx \tilde{\Theta}_L(1),$$

*where $J_1^{iid}$ denotes the Jacobian at initialization (with $\sigma_w^= 2$).*

The result of Theorem 2 suggests that if the Jacobian is stable at initialization, it remains stable throughout training, under approximation Approximation 2. This provides the first theoretical argument for the crucial role of initialization in deep learning. Let us now discuss the three phases of the behaviour of the Jacobian norm during training.

## 4.2 STAGNATION PHASE

One might anticipate that a small learning rate would result in minimal changes to the Jacobian norm, especially at early training. However, it is not straightforward to determine how small the learning rate should be in terms of model characteristics. In the next result, we establish that the Jacobian norm remains asymptotically constant during an initial training stage, the length of which depends on the learning rate.

**Theorem 3.** *Consider a critically initialized MLP (Eq. (1)) trained with the gradient rule given by Eq. (2), with the learning rate satisfying $\gamma < n^{-1}\log(n)^{-3/2}$. Then, for $t = \mathcal{O}(n^{-1}\log(n)^{-3/2}\gamma^{-1})$, under Approximation 1 holding up to time $t$, we have that,*

$$\lim_{n \to \infty} \|J_1^t\| - \|J_1^{iid}\| \approx 0,$$

*where $J_1^{iid}$ denotes the Jacobian with iid weights at initialization.*

Theorem 3 requires that $\gamma < n^{-1}\log(n)^{-3/2}$ for the result to hold. This is a mild condition that is generally satisfied in practice. In fact, the learning rate $\gamma$ should be no larger than $\mathcal{O}(n^{-1})$ otherwise features might explodes as the width grows (Yang and Hu, 2021). We distinguish two different cases depending on the learning rate:

---

[4]Here, the training dataset consists of a single pair $(x, z)$. The analysis can be readily generalized to multiple pairs, but this is an unnecessary complication for this analysis.

[5]Note that here, we don't assume that the weights $W_{ij}^{t'}$ are iid across $(i, j)$. Entries might be correlated.

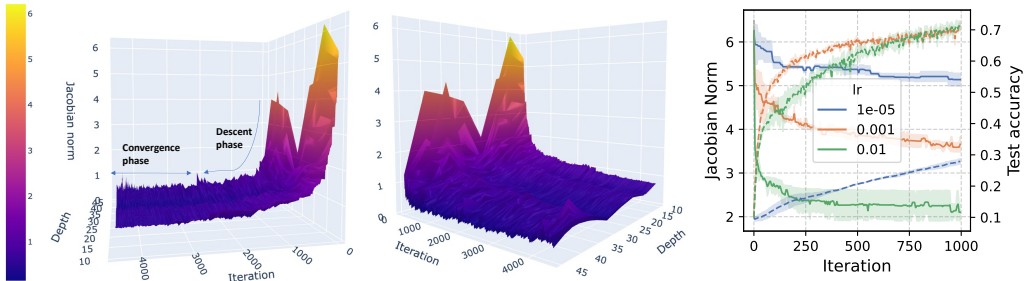

Figure 2: **(Left)** Illustration of the descent and convergence phases on an MLP network with width $n = 256$ and varying depths, trained on CIFAR2 (binary CIFAR10) with stochastic gradient descent. The 2 figures on the left represent the same plot from different angles. We show the first 5000 gradient steps. All the values are averaged over 3 runs. **(Right)** Illustration of the super-lazy training regime Vs the non-lazy regime for an MLP of width $n = 256$ and depth $L = 5$. The dashed, resp. full, lines represent test accuracy, resp. Jacobian norm. We refer the reader to Appendix G for the same figure with larger number of iterations.

- No Feature Learning ($\gamma = \tilde{o}_n(n^{-1})$: the condition on $t$ is satisfied for a large range of $t$ that becomes infinite when $n \to \infty$. As a result, this stagnation phase spans the whole training procedure, suggesting that the Jacobian norm does not change during training. This should be expected since there is no feature learning (Yang and Hu, 2021). Intuitively, the order of magnitude of the learning rate in this case is not sufficient to induce any 'learning' in the infinite-width limit.

- Feature Learning ($\gamma = \tilde{\Theta}_n(n^{-1})$): the condition on $t$ is generally satisfied only at the first few steps (if satisfied at all). The choice $\gamma = \Theta_n(n^{-1})$ is related to the $\mu$-Parametrization which was shown to induce feature learning in the infinite-width limit (Yang and Hu, 2021).[6] Generally, if the learning rate is chosen to be of order $n^{-1}$, the stagnation phase does not appear in this case since feature are quickly updated, inducing a change in the Jacobian norm as well.

We depict in Fig. 2 the effect of small learning rates on the evolution on the Jacobian norm during training. Note that the result of Theorem 3 requires that Approximation 2 holds up to time $t$. We refer the reader to Appendix G for an empirical verification of this approximation.

### 4.3 DESCENT AND CONVERGENCE PHASES

**Descent Phase.** In Fig. 2 (2 figures on the left), the Jacobian norm decreases significantly at early training. In this case, there is no stagnation phase as the learning rate is close to optimal (chosen with a grid search to optimize training, see details in Appendix G). Using a simple toy model, we provide an explanation for the descent behaviour. Consider the Jacobian at time $t$, given by $J_1^t = \prod_{l=2}^{L} W_l^t D_{l-1}^t$. With one gradient step, the Jacobian becomes:

$$J_1^{t+1} = \prod_{l=2}^{L} \left( W_l^t - \gamma \frac{\partial \mathcal{L}}{\partial W_l^t} \right) D_{l-1}^{t+1} = \tilde{J}_1^t - \gamma \Gamma_1^t + \mathcal{O}(\gamma^2),$$

where $\tilde{J}_1^t = \prod_{l=2}^{L} W_l^t D_{l-1}^{t+1}$ and $\Gamma_1^t = \sum_{l'=2}^{L} \left[ \prod_{\substack{l=2 \\ l > l'}}^{L} W_l^t D_{l-1}^{t+1} \right] \frac{\partial \mathcal{L}}{\partial W_{l'}^t} D_{l'-1}^{t+1} \left[ \prod_{\substack{l=2 \\ l < l'}}^{L} W_l^t D_{l-1}^{t+1} \right]$.

The norm of $J_1^{t+1}$ is the square root of the largest eigenvalue of $(J_1^{t+1})^\top J_1^{t+1}$ given by:

$$(J_1^{t+1})^\top J_1^{t+1} = (\tilde{J}_1^t)^\top \tilde{J}_1^t - \gamma((\tilde{J}_1^t)^\top \Gamma_1^t + (\Gamma_1^t)^\top \tilde{J}_1^t) + \mathcal{O}(\gamma^2). \tag{3}$$

A sufficient condition for the largest eigenvalue of $(J_1^{t+1})^\top J_1^{t+1}$ to be smaller than that of $(J_1^t)^\top J_1^t$ is that the first order term in $\gamma$ is non-negative (in matrix sense). A general proof is not straightforward because of the complexity of the training dynamics. However, by restricting our analysis to a simple one-dimensional linear model, we provide insight into this behavior.

**Theorem 4** (Descent with a toy model). *Given a constant $\alpha > 0$, consider the task of learning the (oracle) function $y^*(x) = \alpha x$ using the dataset $\{(x_i, y_i = \alpha x_i)\}_{1 \le i \le N}$ and linear width 1 depth*

---

[6]$\mu$-Parametrization requires some other specific scalings for the initialization and learning rates of the input/output layers.

*L MLP given by $y_{out}(x) = \left[\prod_{l=1}^{L} w_l\right] x$, where $x, w_l, y_{out}(x) \in \mathbb{R}$ (scalars). Then, we have that*

$J_1^t \Gamma_1^t = \bar{x^2} A_t \sum_{l'=1}^{L} \left[\prod_{l \neq l'} w_{l'}^t\right]^2$ *where $A_t = y_{out}^t(1)(y_{out}^t(1) - \alpha)$ and $\bar{x^2} = n^{-1} \sum_{i=1}^{n} x_i^2$. As a result, at initialization, we have $\mathbb{E}_{\mathbf{W}} A_t = 1$, and therefore, on average, $(J_1^t)^2$ is larger than $(J_0^t)^2$ for $t = 1$.*[7]

The proof of Theorem 4 is provided in Appendix D. The result suggests that at early training, we should observe a decent phase in the Jacobian norm. Intuitively, this comes from the fact that gradient-based training always guides the network towards small gradients, thereby forcing the Jacobian (which is involved in the gradients) norm to decrease as well.

**The Convergence Phase.** From Eq. (3), the difference between $(J_1^{t+1})^\top J_1^{t+1}$ and $(\tilde{J}_1^t)^\top \tilde{J}_1^t$ decreases as training progresses (on average). This is attributable to the fact that $\Gamma_1^t$ depends on the gradients, which naturally converge to zero. Consequently, we anticipate the Jacobian norm to approach constancy during training. Our empirical observations corroborate this expectation since the Jacobian norm tends to stabilize in Fig. 2. Surprisingly, this convergence is fast and typically occurs within the first few epochs. The underlying reasons for this phenomenon remain unclear and unexplained at the moment, and further research investigation is necessary to elucidate the factors contributing to this behavior.

To illustrate the *broad applicability of our theoretical framework*, we present another instance of its use in the next section, focusing on sparse networks commonly encountered in the context of network pruning. In Appendix F, we provide another application of our framework for neural networks initialized with non-i.i.d weights.

## 5 JACOBIAN IN SPARSE NETWORKS

Network pruning aims at removing redundant weights that do not significantly affect model performance (LeCun et al., 1990). Such weights are identified using some pruning criterion that determines the importance of each weight in the network. By removing these weights, one could significantly reduce the computational requirements for both the training and deployment of DNNs (Hassibi et al., 1993). After pruning, the network becomes *sparse*, and training such networks has been proven challenging in practice (Frankle and Carbin, 2019b). In this section, we analyze the Jacobian norm of sparse networks and provide the necessary conditions for stability. Notably, we show that a simple method-dependent scaling trick ensures stability. We also identify an *edge of stability* for sparsity and support our theory with empirical evidence. Our current analysis is limited to *pruning at initialization* (Lee et al., 2018b; Wang et al., 2022; Hayou et al., 2021) of MLPs, i.e. pruning performed on i.i.d weights. Extending this theory to modern architectures is not straightforward and is an interesting question for future work.

**Pruning.** Consider the MLP architecture described in Eq. (1). The pruning procedure involves the application of a binary mask $B \in \{0, 1\}^p$ (where $p$ is the total number of parameters in the network, i.e., $p = d \times n + L \times n^2 + o \times n$) to the weights of the network $\mathbf{W}$ producing another network with weights $\mathbf{W}^{pruned}$, where $\mathbf{W}^{pruned} = B \circ \mathbf{W}$ is the Hadamard (i.e., element-wise) product. We say weight $i \in [p]$ is pruned if $b_i = 0$. This can be performed via different procedures. A standard approach to generating masks is to give each weight $W_k^{ij}$ a score $g_k^{ij}$ according to some criterion. The mask is then created by keeping the top $m$ weights by score, where $m$ is chosen to meet some desired sparsity level $s$ (fraction of weights to remove).

To study the stability *after pruning*, we can look at the norm of the Jacobian of the pruned network $J_1^{pruned}$ given by:

$$J_1^{pruned} = \prod_{l=1}^{L} W_l^{pruned} D_{l-1}.$$

We propose to study the Jacobian norm $\|J_1^{pruned}\|$ for networks pruned at initialization with two different pruning methods:

---

[7]Notice that for the toy model, the width is $n = 1$, and therefore, the Jacobian is a real number.

1. **Random Pruning**: weights are randomly pruned with probability $s$ (the sparsity).

2. **Score-Based Pruning**: weights are scored using a certain criteria (e.g. magnitude, sensitivity,).

The first requires a simple application of our main stability theorem (Appendix A), while the latter requires a more delicate stability theorem, the proof of which is deferred to the Appendix E. The main takeaway is that scaling the weights is required to maintain stability in pruned networks. However, the scaling factor depends on the pruning method.

### 5.1 RANDOM PRUNING

In the subsequent analysis, we use the notation $a_n \gg b_n$ for two positive sequences $a_n, b_n$, whenever $b_n = o(a_n)$.

**Theorem 5** (Scaling guarantees stability). *Consider random pruning with sparsity level $s_n \in (0,1)$ that can either depend on $n$ or be constant. Then, under Approximation 1 and the assumption that $1 - s_n \gg \frac{\log(n)^5}{n}$ (in case $s_n$ depends on $n$), by scaling the weights $\mathbf{W}^{pruned}$ with $(1-s_n)^{1/2}$, the Jacobian of the scaled sparse network, given by:*

$$\tilde{J}_1^{pruned} = (1-s_n)^{-L/2} J_1^{pruned} = \prod_{l=1}^{L} (1-s_n)^{-1/2} W_l^{pruned} D_{l-1},$$

*satisfies:*

$$\lim_{n \to \infty} \|\tilde{J}_1^{pruned}\| - \|J_1^{iid}\| \approx 0,$$

*where $J_1^{iid}$ refers to the Jacobian of the non-pruned network with i.i.d weights. As a result, with $\sigma_w^2 = 2$, the pruned network is stable.*

Theorem 5 reveals that, once pruning has been performed, rescaling is essential for stabilizing the Jacobian. Specifically, the theorem states that $\|J_1^{pruned}\|$ is asymptotically proportional to $\tilde{\Theta}_L((1 - s_n)^{(L-1)/2})$. This exponential dependence on depth indicates that stability cannot be attained without modifying the weights. In other words, when starting with a critically initialized network, the weights must be re-scaled after pruning to account for the resultant sparsity. This scaling process renders the infinite-width behavior of the spectral norm similar to that of a non-pruned critically initialized neural network, thereby guaranteeing stability in the sparse network, as evidenced in Fig. 3. Further experiments are provided in Appendix G.

In Theorem 5, the sparsity $s_n$ can depend on $n$. We provide an upper bound on the sparsity in terms of the width, in order for the stability to hold. This result allows us to identify an *edge of stability*, defined as a maximal sparsity (in terms of $n$) so that the stability holds. This result highlights an interesting *phase transition* phenomenon with respect to the sparsity. When the sparsity is of order $1 - s_n \sim n^{-1}$ up to a logarithmic factor, the stability result no longer holds, and the spectral norm of the Jacobian behaves differently (as compared to a non-pruned critically initialed network), see e.g. Benaych-Georges et al. (2019); Tikhomirov and Youssef (2021). In Fig. 3, We observe this behavior when the sparsity hits the level 99%, which is of order $\log_{10}(256)/256 \approx 0.009$. It is worth noting that the condition $1 - s_n \gg n^{-1} \log(n)^5$ is a sufficient condition, and it is highly likely that phase transition occurs at a smaller threshold.

### 5.2 SCORE-BASED PRUNING

To complement the previous section, we show that a similar stability result holds with score-based pruning, however, the scaling constant is different in this case. We restrict our analysis to magnitude-based pruning, performed at initialization, where weights are scored based on their magnitude $g_w = |w|$ Han et al. (2015). However, the proof can be in principle extended to other score-based methods. The main takeaway is that the scaling factor depends on the pruning method.

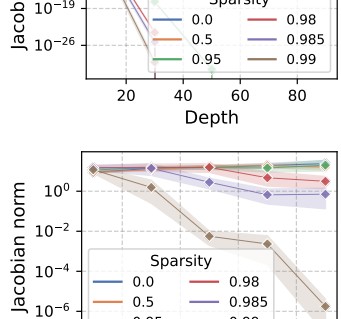

Figure 3: Jacobian norm after pruning at initialization as depth increases in a randomly pruned MLP of width $n = 256$ (input sampled randomly from MNIST). **(Top)** Without scaling. **(Bottom)** With scaling.

**Theorem 6** (Magnitude pruning). *Consider magnitude-based pruning in which the $r_n$ largest weight coefficients in absolute value are kept, corresponding to a sparsity level of $s_n := 1 - \frac{r_n}{n^2}$. Then, under Approximation 1 and the assumption that $1 - s_n \gg \frac{\log(n)^4}{n}$, the Jacobian of the scaled sparse network, given by:*

$$\tilde{J}_1^{pruned} = t_n^L J_1^{pruned} = \prod_{l=1}^{L} t_n W_l^{pruned} D_{l-1},$$

*satisfies:* $\quad \lim_{n \to \infty} \|\tilde{J}_1^{pruned}\| - \|J_1^{ind}\| \approx 0,$
*where $t_n^{-2} := \frac{1}{n^2} \sum_{j=1}^{r_n} \mathbb{E}\Phi^{-1}(1 - B_n^{(j)})$, and, successively, $X$ is a $N(0,1)$ random variable, $\Phi(x) := \mathbb{P}(X^2 \le x)$, and $B_n^{(j)}$ is a $Beta(j, n^2 - j + 1)$ random variable.*

The proof of Theorem 6 is provided in Appendix E.

In Fig. 4, we demonstrate the effect of the scaling factor on the evolution of the Jacobian norm with depth. The top two plots show that without scaling (scaling factor = 1), the Jacobian norm is not stable and remains so when using the same scaling factor as in the case of random pruning $((1 - s_n)^{-\frac{1}{2}})$. The stability is recovered with the correct scaling factor stated in Theorem 6 which is shown in the third plot. The bottom plot compares the two scaling factors, used in the second and third plots respectively, in function of the sparsity. The takeaway is that the optimal scaling factor differs depending on the pruning criteria. The optimal scale for random pruning does not guarantee stability for magnitude-based pruning. It is primordial to determine the correct scaling factor that would guarantee the stability with different pruning criteria. Note that our results on sparse networks confirm previous results on weight scaling in the case of sensitivity-based pruning Hayou et al. (2021)[8].

We refer the reader to Appendix G for results on the performance on trained sparse networks with and without normalization.

## 6 CONCLUSION AND LIMITATIONS

In this paper, we provided an analysis of the Jacobian norm in DNNs in different contexts. Our findings shed light on an important result: stability at initialization implies stability throughout training (under some approximations). In this regard, our work expands the existing literature on this topic and justifies the crucial role of initialization, often observed in practice. We also study the Jacobian of networks pruned at initialization and show how one can stabilize sparse networks after pruning. However, one limitations of our theory is that it currenlty only applies to the MLP architecture. Extending these results to more modern architectures is an interesting question for future work.

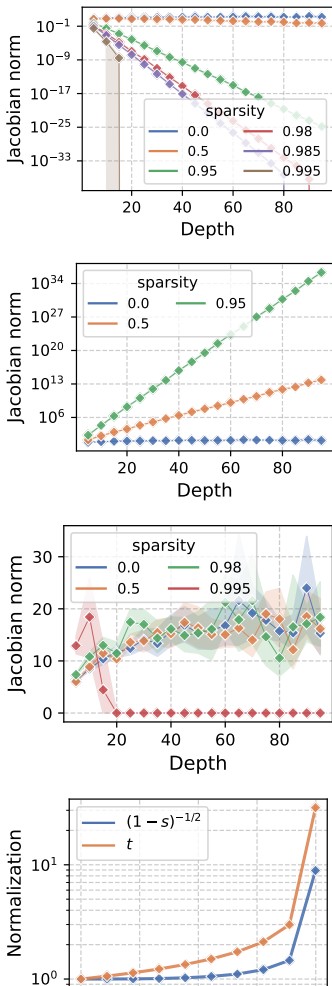

Figure 4: Jacobian norm after pruning at initialization as depth increases in a score-based pruned MLP of width $n = 256$ (input sampled randomly from MNIST). **(Top)** Without scaling (scaling factor = 1). **(Middle)** With scaling: (a) scaling constant $= (1 - s_n)^{-\frac{1}{2}}$; (b) scaling factor $t$ is approximated based on equation E. **(Bottom)** Comparison of the scaling factors in function of sparsity level.

---
[8]It is worth mentioning that the stability measure used in Hayou et al. (2021) is based on the second norm of the gradient, which is a weaker measure than the Jacobian spectral norm.

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

# Appendix

## A  STABILITY THEOREM

Random Matrix Theory (RMT) is a branch of mathematics that deals with the statistical properties of matrices with random entries (Tao, 2011). For a long time, RMT has focused on models with independent and identically distributed entries. However, in recent years, there has been a growing interest in non-homogeneous models, which has led to significant advances in the understanding of the spectral norm of structured random matrices (see Latała et al. (2018) and references therein). Such non-homogeneous models are particularly interesting in applied mathematics as they include sparse matrices and pruned networks. In this work, we rely on recent breakthrough results of Bandeira et al. (2021) which pushes the scope of those obtained in Latała et al. (2018) to cover models with dependent entries. Those advances exploit the connections to the limiting regime of RMT governed by Free Probability Theory (Mingo and Speicher, 2017). This powerful theory uncovers the limiting behavior of the spectrum of a non-commutative polynomial in several independent random matrices. It is therefore well suited to the context of deep neural networks and the spetral study of the Jacobian which is a non-commutative polynomial (a product) of several random matrices (the weight matrices of each layer).

In the sequel, relying on the results of Bandeira et al. (2021), we will derive a general statement capturing the limiting behavior of the input-output Jacobian norm. The generality of this stability theorem will then be exploited to easily derive all statements used in this paper as direct corollaries.

To state the theorem, let

- $W_1, \ldots, W_L$ be independent copies of a $n \times n$ weight matrix $W$ consisting of centered sub-Gaussian coefficients with variance $\mathcal{O}(\frac{1}{n})$ (which may be correlated);

- $D_1, \ldots, D_L$ be independent copies of $D := \mathrm{diag}(d_1, \ldots, d_n)$ with i.i.d. Bernoulli$(\frac{1}{2})$ coefficients $d_i$;

- $B_1, \ldots, B_L$ be independent copies of $tB$ where $t > 0$ and $B := (b_{i,j})$ is a $n \times n$ matrix with uniformly bounded random variables i.e. $\max_{1 \leq i,j \leq n} |b_{i,j}| \leq \beta_n$ for some $\beta_n > 0$.

We also suppose that $(D_i)_{1 \leq i \leq L}$ is independent from $(W_i, B_i)_{1 \leq i \leq L}$. However, we do not require independence between $(B_1, \ldots, B_L)$ and $(W_1, \ldots, W_L)$. Finally, let $J_k := \prod_{l=k}^{L} W_l D_{l-1}$ and $J_k^{pruned} := \prod_{l=k}^{L} B_l \odot W_l D_{l-1}$ (where $\odot$ denotes the Hadamard product).

**Theorem 7** (Stability theorem). *Under the above setting, the convergence*

$$\lim_{n \to \infty} \|J_k^{pruned}\| - \|J_k\| = 0 \tag{4}$$

*holds true if the following three conditions hold:*

*(i)* $\frac{t^2}{n} \log^5 n \, \beta_n^2 \to 0$,

*(ii)* $\sup_{1 \leq i \leq n} \left| t^2 \sum_{k=1}^{n} \mathbb{E}[(b_{i,k} w_{i,k})^2] - 1 \right| \to 0$,

*(iii)* $t^2 n^2 (\log n)^3 \sup_{(i,j) \neq (k,l)} \left| \mathbb{E} b_{i,j} w_{i,j} b_{k,l} w_{k,l} \right| \to 0$.

*Proof.* Let $H := tB \odot WD^9$ and let $C$ denote the $n^2 \times n^2$ matrix with coefficients $C_{(i,j),(k,l)} = \text{cov}(H_{i,j}, H_{k,l})$. According to (Brailovskaya and van Handel, 2022, Theorem 2.16), (4) will hold if

(a) $\ \ \|2\mathbb{E}[HH^*] - \text{Id}_n\| \to 0$,

(b) $\ \ t^2 \, \mathbb{E} \sup_{i,j \leq n} |w_{i,j} b_{i,j}|^2 \to 0$,

(c) $\ \ (\log n)^3 \, \|C\| \to 0$,

and

(d) $\ \ t \log^2 n \sup_{i,j \leq n} |w_{i,j} b_{i,j}| \to 0$ almost surely,

are satisfied, where $\|X\|$ denotes the operator norm (largest singular value) of $X$. We start by checking (b) and (d). By the maximal tail inequality (van Handel, 2016, Lemma 5.2), we have

$$\mathbb{P}\left(\sup_{i,j \leq n} w_{i,j} \geq c\sqrt{\frac{\log n}{n}} + x\right) \leq e^{-Cnx^2}, \quad x > 0$$

for some constants $c, C > 0$. This entails that $\sup_{i,j} |w_{i,j}| \lesssim \sqrt{\frac{\log n}{n}}$ a.s., and $\mathbb{E} \sup_{i,j} w_{i,j}^4 \lesssim \frac{\log^2 n}{n^2}$. Then

$$t^2 \, \mathbb{E} \sup_{i,j \leq n} |w_{i,j} b_{i,j}|^2 \leq t^2 \sqrt{\mathbb{E} \sup_{i,j \leq n} w_{i,j}^4} \sqrt{\mathbb{E} \sup_{i,j \leq n} b_{i,j}^4}$$

$$\lesssim \frac{t^2}{n} \beta_n^2 \log^2 n$$

$$\to 0,$$

by (i), which proves (b). Similarly, (d) holds since, almost surely,

$$\left(t \log^2 n \sup_{i,j \leq n} |w_{i,j} b_{i,j}|\right)^2 \lesssim \frac{t^2}{n} \beta_n^2 \log^5 n \to 0.$$

Next, we have $H_{i,j} = t b_{i,j} w_{i,j} d_j$ and

$$(HH^*)_{i,j} = t^2 \sum_{k=1}^n b_{i,k} w_{i,k} d_k \, b_{j,k} w_{j,k} d_k,$$

so

$$\mathbb{E}(HH^*)_{i,j} = \frac{t^2}{2} \begin{cases} \sum_{k=1}^n \mathbb{E}[b_{i,k} w_{i,k} b_{j,k} w_{j,k}], & \text{if } i \neq j, \\ \sum_{k=1}^n \mathbb{E}(b_{i,k} w_{i,k})^2, & \text{if } i = j. \end{cases}$$

Using the estimate

$$\|A\| \leq \max_{1 \leq k \leq n} \left\{ \sum_{i=1}^n |a_{ik}|, \sum_{j=1}^n |a_{kj}| \right\}$$

for any $n \times n$ matrix $A$ (see, e.g., (Qi, 1984, Theorem 2)), it is then easy to check that

$$\|2\mathbb{E}[HH^*] - \text{Id}_n\| \leq \sup_{1 \leq i \leq n} \left| t^2 \sum_{k=1}^n \mathbb{E}[(b_{i,k} w_{i,k})^2] - 1 \right| + t^2 n^2 \sup_{(i,j) \neq (k,l)} |\mathbb{E}[b_{i,j} w_{i,j} b_{k,l} w_{k,l}]|.$$

---

[9] In (Brailovskaya and van Handel, 2022, Theorem 2.16), the authors consider self-adjoint matrices, but the result holds for any matrix as they pointed in Remark 2.1 in the same paper. This is straight-forward using the hermitization trick: given a non-Hermitian matrix $M$, define $H = \begin{bmatrix} 0 & M \\ M^* & 0 \end{bmatrix}$ which is Hermitian and its eigenvalues are $\pm$ the singular values of $M$.

Therefore, (a) is satisfied under (ii) and (iii). To check (c), we have

$$C_{(i,j),(k,l)} = \mathbb{E}[H_{i,j}H_{k,l}] = t^2 \begin{cases} \mathbb{E}[b_{i,j}b_{k,l}w_{i,j}w_{k,l}]\frac{1+\delta_{j,l}}{4}, & \text{if } (i,j) \neq (k,l), \\ \frac{1}{2}\mathbb{E}(b_{i,j}w_{i,j})^2, & \text{if } (i,j) = (k,l), \end{cases}$$

Then, for any $n \times n$ matrix $M$,

$$\left| \sum_{i,j,k,l} M_{i,j} C_{(i,j),(k,l)} M_{k,l} \right| = \frac{t^2}{2}\sum_{i,j}\mathbb{E}(b_{i,j}w_{i,j})^2 M_{i,j}^2 + \frac{t^2}{4}\sum_{(i,j)\neq(k,l)}\mathbb{E}[b_{i,j}w_{i,j}b_{k,l}w_{k,l}]M_{i,j}M_{k,l}(1+\delta_{j,l})$$

$$\leq \frac{t^2}{2}\left(\frac{\beta_n^2}{n} + n^2 \sup_{(i,j)\neq(k,l)}\left|\mathbb{E}b_{i,j}w_{i,j}b_{k,l}w_{k,l}\right|\right)\sum_{i,j}M_{i,j}^2,$$

so (c) holds thanks to (i) and (iii). $\qquad\square$

## B  PROOF OF THEOREM 2

*Proof.* For some $l_2 < l_1$, denote $J_1^{l_1:l_2} = \prod_{l_1 \geq l \geq l_2} W_l^t D_{l-1}^{t+1}$.

Recall that

$$J_1^{t+1} = \prod_{l=1}^{L}\left(W_l^t - \gamma\frac{\partial\mathcal{L}}{\partial W_l^t}\right)D_{l-1}^{t+1} = \tilde{J}_1^t + \sum_{k=1}^{L}(-\gamma)^k\Gamma_k^t.$$

where, $\tilde{J}_1^t = \prod_{l=2}^{L} W_l^t D_{l-1}^{t+1}$ and,

$$\Gamma_k^t = \sum_{L\geq l_1>l_2>\cdots>l_k\geq 1} J_1^{L:l_1+1}\frac{\partial\mathcal{L}}{\partial W_{l_1}^t}D_{l_1-1}^{t+1}J_1^{l_1-1:l_2+1}\frac{\partial\mathcal{L}}{\partial W_{l_2}^t}D_{l_2-1}^{t+1}\cdots\frac{\partial\mathcal{L}}{\partial W_{l_k}^t}D_{l_k-1}^{t+1}J_1^{l_k-1:1}.$$

For instance, $\Gamma_L^t = \prod_{l=1}^{L}\frac{\partial\mathcal{L}}{\partial W_l^t}$ and $\Gamma_1^t = \sum_{l'=1}^{L}\left[\prod_{l>l'}^{L} W_l^t D_{l-1}^{t+1}\right]\frac{\partial\mathcal{L}}{\partial W_{l'}^t}D_{l'-1}^{t+1}\left[\prod_{l<l'}^{L} W_l^t D_{l-1}^{t+1}\right].$

The key ingredient in this proof is to prove that under Approximation 2 holding up to step $t$, the terms in $\Gamma_k^t$ are all of order $\Theta_L(L^{2k+1})$. To show this, we proceed by induction and use the fact that for iid initialized weights, we have (Theorem 1) $\|J_1\| \lesssim CL$ for some constant $C$ (that potentially depends on the input $x$ but not on depth $L$). Observe also that $\frac{\partial\mathcal{L}}{\partial W_l^t} = J^{L:l+1}dY_L^t\phi(Y_{l-1}^t)^\top$, where $dY_L^t = \frac{\partial\mathcal{L}}{\partial Y_L}$ at step $t$. Hence, for $t = 1$, we have that

$$\|\Gamma_k^t\| \lesssim \sum_{L\geq l_1>l_2>\cdots>l_k\geq 1}(CL)^{2k+1} \times \prod_{j=1}^{k}\|dY_L^{t=0}\phi(Y_{l_j-1}^{t=0})^\top\|.$$

Furthermore, we have that $\|dY_L^{t=0}\phi(Y_{l-1}^{t=0})^\top\| = \|\phi(Y_{l-1}^{t=0})\|_e\|dY_L^{t=0}\|_e$, where $\|.\|_e$ refers to the Euclidean norm in $\mathbb{R}^n$. By assumption, we have $dY_L^{t=0} = n^{-1}(Y_L^{t=0} - z)$ and therefore $\lim_{n\to\infty}\|dY_L^{t=0}\|_e = \Theta_L(1)$.[10] Similarly, we have that $\lim_{n\to\infty}n^{-1}\|\phi(Y_{l-1}^{t=0})\|_e^2 = \Theta_L(1)$. This follows also from the same argument above. More generally, the previous two bounds hold throughout training using the Tensor Program framework Yang and Hu (2021) (Thm 7.4). Therefore, letting $\tilde{\gamma} = \lim_{n\to\infty}n\gamma$ (which exists by assumption), we obtain

$$\|J_1^1\| \lesssim CL + L\sum_{k=1}^{L}\binom{L}{k}(C\tilde{\gamma}L^2)^k = \Theta_L(L \times (1 + C\tilde{\gamma}L^2)^L).$$

Therefore if $\gamma = \mathcal{O}(n^{-1}L^{-3})$, we have $\|J_1^1\| \approx \Theta_L(1)$.

Using the "Master Theorem" from Yang and Hu (2021), the concentration argument mentioned above holds for any $t$, more precisely, we have $\lim_{n\to\infty}\|dY_L^{t=t'}\|_e = \Theta_L(1)$ and $n^{-1}\lim_{n\to\infty}\|\phi(Y_{l-1}^{t=t'})\|_e^2 = \Theta_L(1)$. We conclude that this is true for all any $t$, provided that Approximation 2 holds up to time $t$.

$\qquad\square$

---

[10]This uses standard concentration results on the neurons in an MLP, see e.g. Matthews et al. (2018).

## C    PROOF OF THEOREM 3 AND THEOREM 5

The setting of both theorems corresponds to letting the mask $B$ be random and independent of the weight matrix $W$, where the entries of $B$ are i.i.d Bernoulli variables with parameter $1 - s$ (note that $s = 0$ in the case of Theorem 3). In this case $\mathbb{E}[b_{i,j}] = 1 - s$, and the conditions of the stability theorem reduce to $t \sim (1 - s)^{-\frac{1}{2}}$, $1 - s \gg \frac{\log^5 n}{n}$, and $(1 - s)\gamma^2 n^2 (\log n)^3 \to 0$ (note that $\gamma = 0$ in the case of Theorem 5). This proves both Theorem 5 and Theorem 3. For Theorem 3, it should be noted that we use the fact that if the Jacobian is stable at initilization, then it remains stable thourghout traingning under Approximation 2 (Theorem 2). This combined with the condition on the learning rate and that of $t$ yields the desired result.

## D    PROOF OF THEOREM 4

Given a constant $\alpha > 0$, consider the task of learning the (oracle) function $y^*(x) = \alpha x$ using the dataset $\{(x_i, y_i = \alpha x_i)\}_{1 \leq i \leq N}$. The model consists of a depth $L$ linear MLP given by $y_{out}(x) = \left[\prod_{l=1}^{L} w_l\right] x$, where $x, w_l, y_{out}(x) \in \mathbb{R}$ (scalars)[11]. With the mean squared loss, simple calculations yields $\frac{\partial \mathcal{L}}{\partial w_l} = \left[\prod_{l' \neq l} w_{l'}\right] \bar{x^2}(y_{out}(1) - \alpha)$, where $\bar{x^2} \overset{def}{=} \frac{1}{n} \sum_{i=1}^{N} x_i^2$. The Jacobian at training step $t$ is a scalar and is equal to $J_1^t = \prod_{l=1}^{L} w_l^t$, where $w_l^t$ is given by the gradient rule $w_l^{t+1} = w_l^t - \gamma \frac{\partial \mathcal{L}}{\partial w_l^t}$. With these dynamics, Eq. (3) becomes

$$(J_1^{t+1})^2 = (J_1^t)^2 - 2\gamma J_1^t \Gamma_1^t + \mathcal{O}(\gamma^2),$$

where $\Gamma_1^t = \sum_{l'=1}^{L} \left[\prod_{l \neq l'} w_{l'}^t\right] \frac{\partial \mathcal{L}}{\partial w_{l'}^t} = \bar{x^2}(y_{out}^t(1) - \alpha) \sum_{l'=1}^{L} \left[\prod_{l \neq l'} w_{l'}^t\right]^2$, where $y_{out}^t$ refers to the network output with parameters $w^t$. Therefore, we obtain

$$J_1^t \Gamma_1^t = \bar{x^2} y_{out}^t(1)(y_{out}^t(1) - \alpha) \sum_{l'=1}^{L} \left[\prod_{l \neq l'} w_{l'}^t\right]^2 .$$

The term $\sum_{l'=1}^{L} \left[\prod_{l \neq l'} w_{l'}^t\right]^2$ is positive almost surely. To conclude, let us look at the average behavior of the term $y_{out}^t(1)(y_{out}^t(1) - \alpha)$ at early training stages. At step $t = 0$ (init), we have that $\mathbb{E}_{\mathbf{W}}\left[y_{out}^0(1)(y_{out}^0(1) - \alpha)\right] = \mathbb{E}_{\mathbf{W}}\left[y_{out}^0(1)^2\right] = 1$, and therefore on average $(J_1^1)^2$ is smaller than $(J_1^0)^2$. Intuitively, this behavior should persist during the first few steps.

## E    PROOF OF THEOREM 6

In the score-based pruning, we compute some scores $g \in \mathbb{R}^p$ and prune based on the value of $g$ (keep the $(1 - s) \times p$ weights with the highest scores). In this case, the weights $W_l$ are dependent, but this dependence is weak when the number of parameters $p$ is large because in this case, score-based pruning becomes close to a rejection algorithm.

In the setting of Theorem 6, the magnitude-based pruning consists in choosing $b_{i,j} := \mathbf{1}_{\{|w_{i,j}| \geq w^{(r)}\}}$ for the mask, where $w^{(1)} \geq w^{(2)} \geq \cdots \geq w^{(n^2)}$ is the non-increasing rearrangement of $\{|w_{i,j}| : i, j \leq n\}$. Moreover, we suppose that the gaussian weights are uncorrelated. To apply the stability theorem, we estimate

$$\mathbb{E}(b_{i,k} w_{i,k})^2 = \mathbb{E}w_{i,k}^2 \mathbf{1}_{\{|w_{i,k}| \geq w^{(r)}\}}$$

$$= \frac{1}{n} \sum_{j=1}^{r} \mathbb{E}\left(X^2 \cdot \binom{n^2 - 1}{j - 1} \Phi(X^2)^{n^2 - j}\left(1 - \Phi(X^2)\right)^{j-1}\right),$$

$$= \frac{1}{n^3} \sum_{j=1}^{r} \mathbb{E}\Phi^{-1}(1 - B_n^{(j)}),$$

---

[11]It is worth noting that this simple model excludes the weights $w_{out}$ and $w_{in}$. Nevertheless, the inclusion of these weights does not affect the validity of the result.

where, successively, $X$ is a $N(0,1)$ random variable, $\Phi(x) := \mathbb{P}(X^2 \leq x)$, and $B_n^{(j)}$ is a Beta$(j, n^2 - j + 1)$ random variable. Thus, we set

$$t^{-2} := \sum_{k=1}^n \mathbb{E}(b_{i,k} w_{i,k})^2 = \frac{1}{n^2} \sum_{j=1}^r \mathbb{E}\Phi^{-1}(1 - B_n^{(j)}) \tag{5}$$

to enforce condition (ii) of the stability theorem. Note that

$$\Phi^{-1}(x) = 2\operatorname{erf}^{-1}(x)^2$$

where $\operatorname{erf}^{-1}$, the inverse of the standard error function, fulfills

$$\operatorname{erf}^{-1}(1-x)^2 \leq \log \frac{1}{x}, \quad x \in (0, 1],$$

$$\operatorname{erf}^{-1}(1-x)^2 \sim \log \frac{1}{x}, \quad x \to 0.$$

For all $j \ll n^2$, the law of large numbers shows that $B_n^{(j)} \sim \frac{j}{n^2}$ almost surely as $n \to \infty$. In particular,

$$\mathbb{E}\left( \frac{\operatorname{erf}^{-1}\left(1 - B_n^{(j)}\right)^2}{\log \frac{n^2}{j}} \right)^2 \lesssim \frac{\mathbb{E} \log^2 B_n^{(j)}}{\log^2 \frac{n^2}{j}} = O(1),$$

where

$$\frac{\operatorname{erf}^{-1}\left(1 - B_n^{(j)}\right)^2}{\log \frac{n^2}{j}} \to 1.$$

It follows by uniform integrability that

$$\mathbb{E}\Phi^{-1}(1 - B_n^{(j)}) \sim 2\log \frac{n^2}{j}$$

whenever $j \ll n^2$. Thus, if we choose $r \gg n \log^4 n$, then for all $1 \leq j \leq \lceil \sqrt{rn} \log^2 n \rceil =: R$,

$$\mathbb{E}\Phi^{-1}(1 - B_n^{(j)}) \geq \Phi^{-1}(1 - B_n^{(R)}) \sim 2\log \frac{n^2}{R} \asymp \log n$$

so

$$t^2 \leq \frac{n^2}{\sum_{j=1}^R \mathbb{E}\Phi^{-1}(1 - B_n^{(j)})} \lesssim \frac{n^2}{R \log n} \sim \frac{n^2}{\sqrt{rn} \log^3 n} \ll \frac{n}{\log^5 n},$$

which shows that condition (i) of the stability theorem holds. Condition (iii) is null since

$$\mathbb{E}b_{i,j} w_{i,j} b_{k,l} w_{k,l} = \mathbb{E}w_{i,j} w_{k,l} \mathbf{1}_{\{|w_{i,j}| \wedge |w^{(k,l)}| \geq w^{(r)}\}} = 0,$$

by symmetry. Thus stability will hold for the magnitude-based pruning if we keep e.g. the $r = \lceil n \log^5 n \rceil$ largest coefficients of the weight matrix. This proves Theorem 6.

## F  FURTHER THEORETICAL RESULTS: JACOBIAN WITH DEPENDENT WEIGHTS

The reader might ask what happens to the Jacobian norm when the weights are *not* independent. Another way of posing this question is as follows:
*starting from a critical initialization as described in Theorem 1, what level of correlation may be introduced between the weights without jeopardizing the network's stability?*

The next theorem demonstrates that weight matrices $\mathbf{W}$ that possess correlated entries can result in network stability, provided that the degree of correlation does not surpass $o(n^{-1} \log(n)^{-3})$. In this case, we also necessitate that Approximation 1 holds, and a rationale for this requirement is provided subsequent to the statement of the theorem. The proof of the theorem is deferred to **??**.

**Theorem 8** (Stability with Dependent Weights). *Assume that the weights $W_1, W_2, \ldots, W_L$ are independent copies of $n \times n$ weight matrix $W$ consisting of centered Gaussian entries with variance $\sigma_w^2/n$ and correlation $\mathrm{corr}(W_k^{ij}, W_k^{ml}) = \mathcal{O}(n^{-1} \log(n)^{-3})$ if $(i,j) \neq (m,l)$. Then, under Approximation 1, in the limit $n \to \infty$, we have the following:*

$$\lim_{n \to \infty} \|J_1\| - \|J_1^{iid}\| \approx 0,$$

*where $J_1^{iid}$ denotes the Jacobian with iid weights at initialization. As a result, the stability holds in this case also with $\sigma_w^2 = 2$.*

*Proof.* It suffices to apply the stability theorem with $t = 1$ and the (trivial) mask $B$ where all components are equal to $1$ (no pruning). In the setting of Theorem 8 the three conditions of the stability reduce to $\mathrm{cov}(W_k^{ij}, W_k^{ml}) = o(n^{-2} \log(n)^{-3})$, that is $\mathrm{corr}(W_k^{ij}, W_k^{ml}) = o(n^{-1} \log(n)^{-3})$. $\qquad\square$

It is reasonable to anticipate that when the correlation between weights is small enough, the Jacobian norm will display behavior akin to that observed in the i.i.d case. However, the task of quantifying the degree of correlation required for such similarity to hold is not straightforward. Theorem 8 provides a necessary condition on the correlation, in terms of a comparison between the correlation and the width $n$. Provided that the correlation is much smaller than $n^{-1}$ (up to a logarithmic term), the Jacobian norm will be virtually identical to its i.i.d counterpart. An empirical verification of Approximation 1 for correlated weights is provided in Appendix G. In Fig. 5, we demonstrate the influence of depth and the correlation between weights on the Jacobian norm. In our simulations, the weights are generated as $W_k^{ij} = W_{k,ind}^{ij} + \eta w^k$, where $w^k, W_{k,ind}^{ij} \sim \mathcal{N}(0, 2/n)$, and $\eta$ is held constant. In this setup, we have $\mathrm{corr}(W_k^{ij}, W_k^{ml}) = \frac{\eta^2}{1+\eta^2} \approx \eta^2$ when $\eta$ is small. For $n = 256 = 2^8$, the condition specified in Theorem 8 translates to $\eta \ll 2^{-4}$. As seen in the figure, for $\eta \in \{2^{-8}, 2^{-7}\}$, the Jacobian norm closely matches the i.i.d case (represented by the blue curve), particularly when $L \leq 40$. As depth increases, one would expect that the difference between the correlated and the i.i.d Jacobian norms to become more pronounced. This is due to the fact that, given a fixed depth, the result is valid in the infinite-width limit[12].

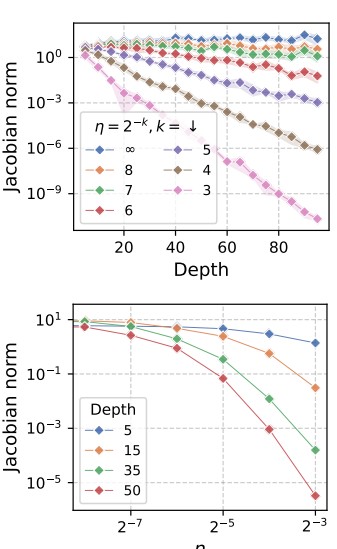

Figure 5: Illustration of the Jacobian norm for a randomly selected input with an MLP architecture of width $n = 256$ and varying depths. All results are averaged over 3 runs, and confidence intervals are highlighted with shaded areas. **(Top)** Impact of Depth on the Jacobian norm for different correlation levels. **(Bottom)** Impact of the injection of the correlation between the weights on the Jacobian norm.

## G  Further empirical results

In this section, we conduct several experiments to verify the approximations used in the main text. We also train different architectures (MLP, VGG) to confirm some theoretical results. All the networks are trained with SGD, and learning rate is tuned using a grid search in the set $\{1e-1, 1e-2, 1e-3, 1e-4\}$. For VGG network, this reflects the initial learning rate (decreased twice during training).

### G.1  Empirical verification of Approximation 1

In this section, we empirically verify Approximation 1 using a mixture of independence test and visualization methods. We depict the results in three cases: the i.i.d weights case, the dependent weights case, and the super-lazy training case.

---

[12]Note that if depth = width and both tend to infinity, Theorem 6 and Theorem 1 do not hold.

### G.1.1 IID WEIGHTS

**Diagonal entries of $D_l$.** We first verify that the diagonal entries of $D_l$ are approximately iid Bernoulli variables with parameter $1/2$.

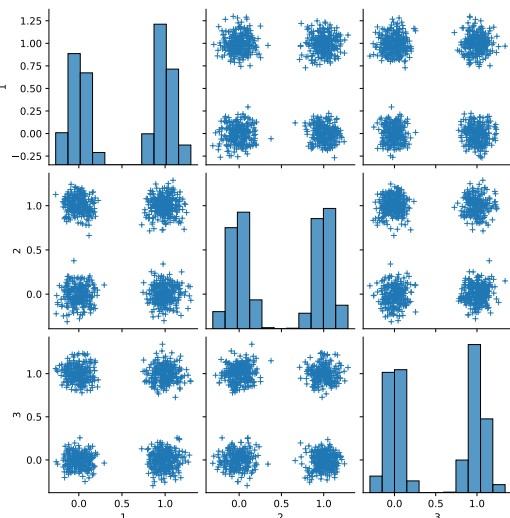

Figure 6: Joint distributions of three randomly selected entries of $D_l$ (denoted by 1, 2, and 3) for $l = 10$ in a depth $L = 30$ and width $n = 100$ MLP with a randomly selected input, based on $N = 1000$ simulations. Since the values of the entries are binary (0 or 1) we added random Gaussian noise (variance 0.01) to the points for better visibility.

In Fig. 6, we show the joint distributions of three randomly selected entries of $D_l$ for $l = 10$ in a depth $L = 30$ and width $n = 100$ MLP. Since the values of the entries are binary (0 or 1) we added random Gaussian noise (variance 0.01) to the points for better visibility. The two main observations are the absence of correlation between the values of the entries, and that each entry has approximately probability $1/2$ of being equal to 1, thus confirming the validity of Approximation 1 in the iid weights case.

*Chi-squared independence test.* We further run a chi-squared independence test between two randomly selected entries of $D_l$ and we obtained the following results: $\chi^2(1) = 0.1379$, and p-value $= 0.73$, and thus the "H0" hypothesis (independent random variables) cannot be rejected. Another observation is that the p-value seems to have a uniform distribution as we change the random seed, which further supports the independence hypothesis[13].

**Independence of $W_l$ and $D_l$.** There is no standard method to evaluate the independence between a discrete and a continuous random variable. We use the following heuristic to evaluate dependence between the matrices $W_l$ and $B_l$: we compute the statistics $T_W = \frac{1}{n^2} \sum_{1 \leq i,j \leq n} 1_{W_l^{ij} > 0}$ and $T_D = \frac{1}{n} \sum_{1 \leq i \leq n} 1_{D_l^i > 0}$ and study the correlation between them.

Fig. 7 shows the joint distribution of $(T_W, T_D)$ for $D_l$ and $W_l$ ($l = 10$) in a $L = 30$ and $n = 100$ MLP. No clear correlation can be observed from the histograms. This supports the approximation of independence between $W$ and $D$. We will see in the case of dependent weights that this statistics become correlated when we increase the correlation level, confirming the validity of this simple heuristic.

### G.1.2 DEPENDENT WEIGHTS

**Diagonal entries of $D_l$.** Fig. 8 shows the joint distributions of three randomly selected diagonal entries of $D$ as in the previous selection with the only difference being that the weights are now

---

[13]It is well known that the distribution of the p-value under the H0 hypothesis is uniform in $[0, 1]$

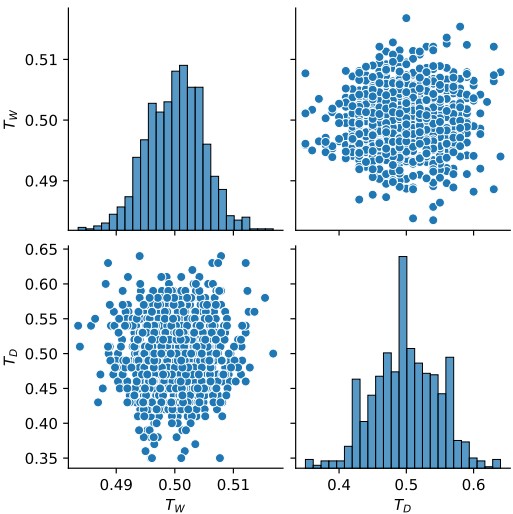

Figure 7: Joint distribution of $(T_W, T_D)$ for $D_l$ and $W_l$ ($l = 10$) in a $L = 30$ and $n = 100$ MLP with a randomly selected input, based on $N = 1000$ simulation.

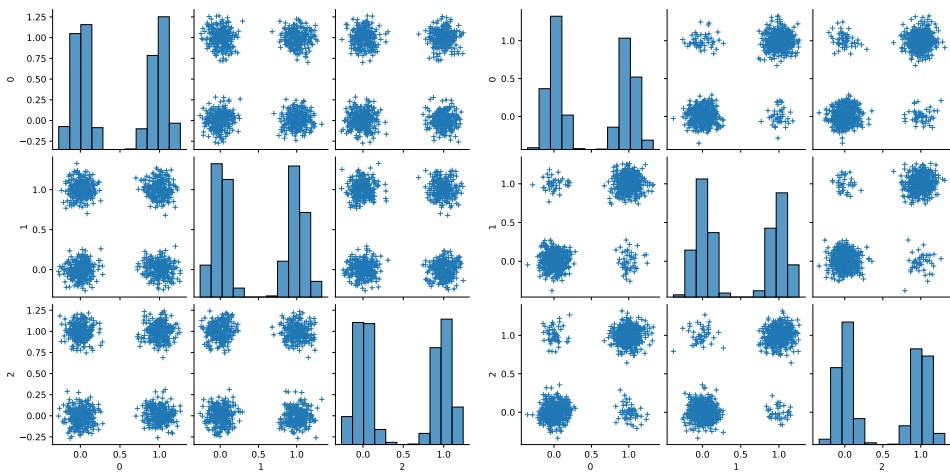

Figure 8: Same as Fig. 6 with dependent weights given by $\tilde{W}_l^{ij} = W_l^{ij} + \eta w$ as in Fig. 5. **(Left)** $\eta = 0.1$. **(Right)** $\eta = 0.9$.

correlated. With weak correlation (left figure), it appears that Approximation 1 is still a good approximation. However, with high correlation (right figure), it is clear that the diagonal entries are not dependent and the approximation is not valid in this case.

*Chi-squared independence test.* (Left) $\chi^2(1) = 0.11$, and p-value $= 0.45$, (Right) $\chi^2(1) = 61.01$, and p-value $= 1e - 6$. Therefore, the independence hypothesis cannot be rejected in the low correlation case, while it is (strongly) rejected in the high correlation case.

**Independence of $W_l$ and $D_l$.** In Fig. 9, we use the same heuristic as in Fig. 7 for both correlation levels $\eta = 0.1$ and $\eta = 0.9$. While the first case no clear correlation can be observed between $T_W$ and $T_D$, it is straightforward from the figure on the right that the two statistics are dependent, suggesting that $W_l$ and $D_l$ are dependent in this case. This confirms the validity of our heuristic in studying the independent between $W_l$ and $D_l$.

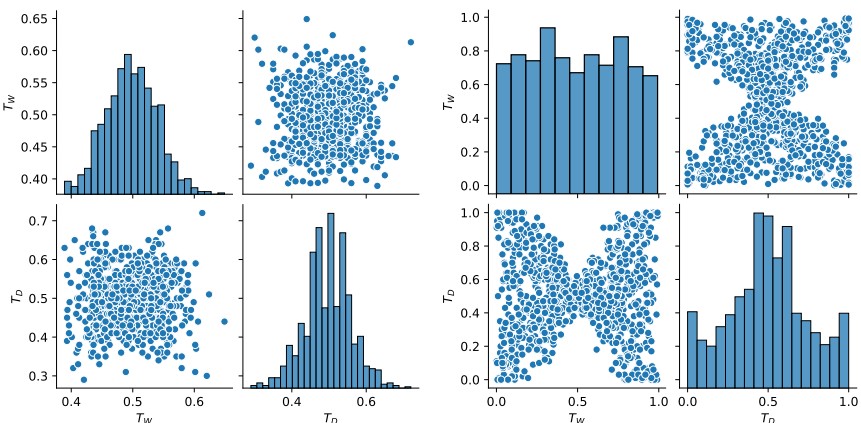

Figure 9: Joint distribution of $(T_W, T_D)$ for $D_l$ and $W_l$ ($l = 10$) in a $L = 30$ and $n = 100$ MLP with a randomly selected input, based on $N = 1000$ simulation. **(Left)** $\eta = 0.1$. **(Right)** $\eta = 0.9$.

## G.2 Approximation 2

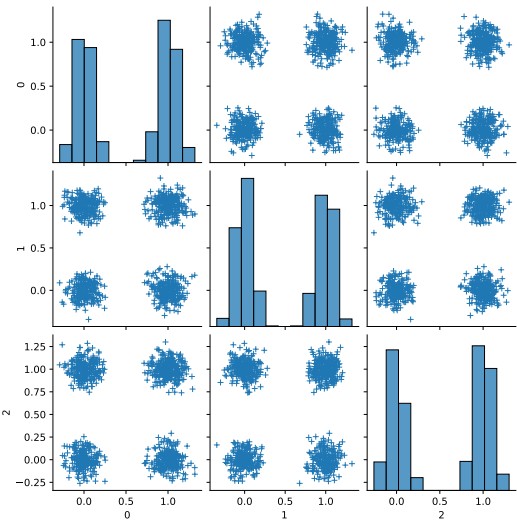

Figure 10: Same as Fig. 6 with trained weights (iteration 4000) with learning rate $\gamma = 1e - 4$.

**Diagonal entries of $D_l$.** Fig. 10 shows the joint distributions of three randomly selected diagonal entries of $D$ as in the previous selection with the only difference being that the weights are now the trained weights given by $\mathbf{W}^t$ at iteration $t = 4000$ with learning rate $\gamma = 1e-4$. Up to this iteration, it appears that Approximation 2 is still a good approximation. It should be expected however that as the number of iterations grows, the independence approximation should no longer be viable.

**Independence of $W_l$ and $D_l$.** In Fig. 11, we use the same heuristic as in Fig. 7 for trained weights at iteration $t = 4000$. No clear correlation can be observed between $T_W$ and $T_D$ from the figure.

## G.3 Different Learning Rates

Fig. 12 is the same as Fig. 2 (Right), with more training steps.

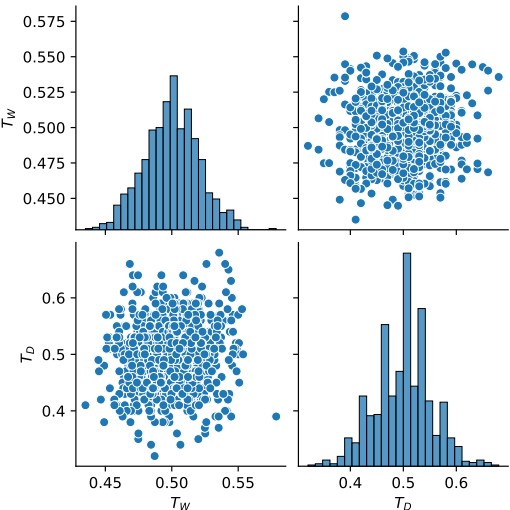

Figure 11: Joint distribution of $(T_W, T_D)$ for $D_l$ and $W_l$ ($l = 10$) in a $L = 30$ and $n = 100$ MLP with a randomly selected input, based on $N = 1000$ simulations of the training procedure. Iteration $t = 4000$.

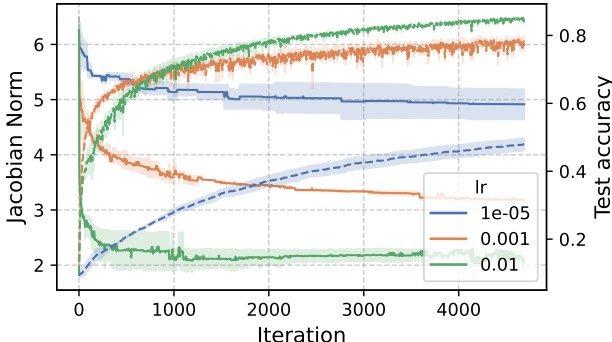

Figure 12: Jacobian+TestAcc evolution in a MLP of width $n = 256$ and depth $L = 5$ for different learning rates. The dashed, resp. full, lines represent test accuracy, resp. Jacobian norm.

### G.4    IMPACT OF NORMALIZATION IN PRUNED NETWORKS

In Fig. 13 and Fig. 14, we report the test accuracy/loss of an MLP of width $n = 256$ of varying depths trained on Fashion-MNIST. The normalization procedure significantly improves the trainability of the network after pruning. The edge of stability can also be observed in terms of trainability as predicted in Theorem 5.

### G.5    FULL NETWORK WITH I.I.D AND DEPENDENT WEIGHTS

In Fig. 15, we depict the test error after convergence in two settings: IID initialized MLP trained on MNIST, and Dependent Weights Initialized MLP trained on Fashion-MNIST. The results support our theoretical findings.

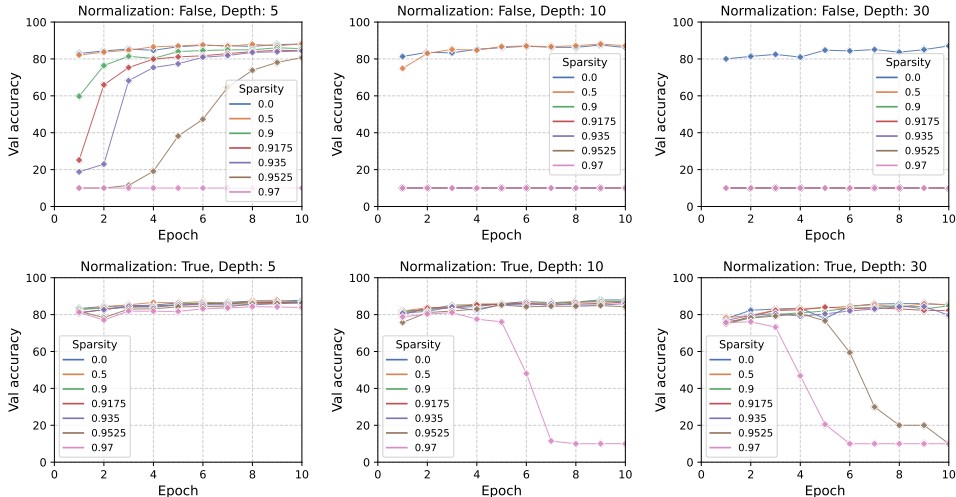

Figure 13: Test (Validation) accuracy of a width $n = 256$ *randomly pruned* MLP of varying depths trained on Fashion-MNIST with learning rate $\gamma = 0.01$. Top figures show the results without normalization while the bottom figures show the results with normalization.

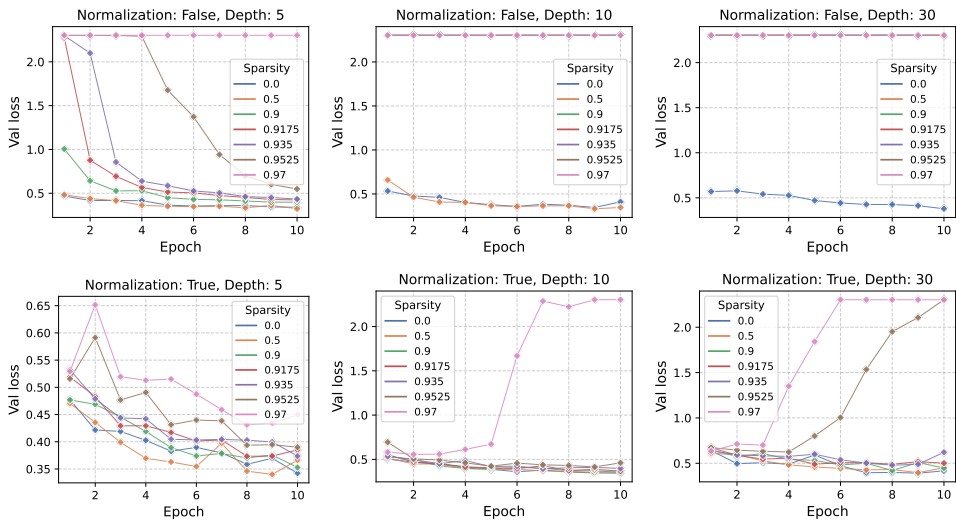

Figure 14: Test loss of a width $n = 256$ *randomly pruned* MLP of varying depths trained on Fashion-MNIST with learning rate $\gamma = 0.01$. Top figures show the results without normalization while the bottom figures show the results with normalization.

## G.6 EXPERIMENTS WITH VGG ARCHITECTURES

We conducted additional experiments with VGG networks (CNN) trained on CIFAR10 dataset. The weights are initialized as zero-mean Gaussian variables with variance $\sigma_w^2/(c(2k+1))$ where $c$ is the fan-in number of channels, $k$ is the filter size (fixed to 3), and $\sigma_w$ is a hyper-parameter that we vary to obtain different values of the Jacobian norm (see Xiao et al. (2018) for an explanation of this choice of the variance). The convolutional layers have a padding step of 1 and the padding is circular. We consider VGG archs of depths $L \in [5, 15, 25, 35]$. The number of pooling layers is fixed to 5 and does not increase with depth. We refer to the pooling layer by 'MP' and the last dense layer by 'FC' (Fully connected). The architectures are given by (we count the FC layer in the depth):

Depth $L = 5$ : [64, 'MP', 128, 'MP', 256, 'MP', '512', 'MP', 'FC' ]

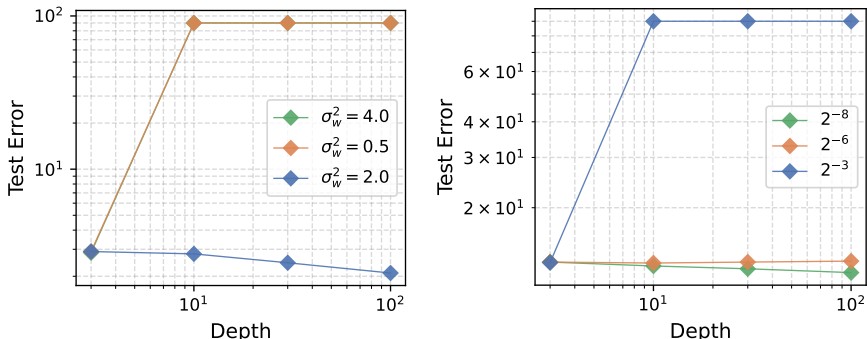

Figure 15: **(Left)** Test error after convergence (70 epochs) as a function of depth in an MLP with width $n = 256$ trained with SGD with learning rate $\gamma = 0.001$ on MNIST. The results are shown for 3 different choices of $\sigma_w$, the variance of the iid weights. The critical initialization given by $\sigma_w^2 = 2$ guarantees trainability up to depth 100. Note that the orange and green curves coincide, hence the green curve is not visible. **(Right)** Test error after convergence (70 epochs) as a function of depth in an MLP with width $n = 256$ trained with SGD with learning rate $\gamma = 0.001$ on Fashion-MNIST. The results are shown for 3 different choices of the correlation parameter $\eta$. When the correlation is low, the network remains trainable for large depths, which is not the case for high correlation levels.

Depth $L = 15$ : [[64]*2, 'MP', [128]*4, 'MP', [256]*4, 'MP', ['512']*4, 'MP', 'FC' ]

Depth $L = 25$ : [[64]*3, 'MP', [128]*7, 'MP', [256]*7, 'MP', [512]*7, 'MP', 'FC' ]

Depth $L = 35$ : [[64]*4, 'MP', [128]*10, 'MP', [256]*10, 'MP', [512]*10, 'MP', 'FC' ]

All the networks are trained with SGD (details in Fig 16 caption).

Fig 16: Train/Test accuracy for different depths with/without BatchNorm(BN). BN seems to improve performance when Jacobian norm is large, but fails when the latter is extremely large (of order 1e8). When the Jacobian norm is well-behaved, the network is trainable without BN, however, it seems that BN improves test accuracy by 2

Fig 17: We vary the hyperparameter to obtain different values of the Jacobian norm at initialization. On the right figure, we show the Test Acc VS Jacobian Norm (left). The results further support the association between good properties of Jacobian and trainability/performance. BN seems to mitigate Jacobian exploding/vanishing up to a certain limit where training with BN also fails. The figure on the left shows the TrainingTime VS Jacobian Norm. It can be inferred that unstable Jacobian norms are associated with longer training times (nb of epochs until convergence).

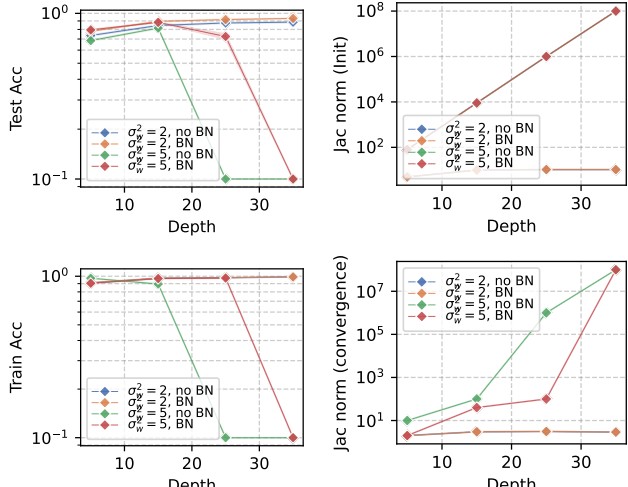

Figure 16: **(Setup)** VGG trained on CIFAR10 dataset with SGD. The initial learning rate is divided by $10$ after the first $50$ epochs, then divided a second time by $10$ after $25$ epochs if convergence is not attained. The initial learning rate is fixed to $0.1$ (found to be optimal by a logarithmic scale grid search). Training is stopped when the training loss (cross-entropy) does not significantly change for $5$ consecutive epochs (remains within 1e-4 bound for $5$ consecutive epochs). **(Left)** Train/Test error after convergence as a function of depth. The results are shown for $2$ different choices of $\sigma_w$ (the variance of the init weights) and whether we use BatchNorm or not. The critical initialization given by $\sigma_w^2 = 2$ guarantees trainability up to depth $L = 35$. The use of BN in this case does not seem to improve training error (with $\sigma_w = 2$) but it improves the test accuracy. However, when the Jacobian norm is large ($\sigma_w^2 = 5$), BN improves trainability for $L = 25$, but still underperforms stable training ($\sigma_w^2 = 2$) without BN. For depth $L = 35$, even with BN, we were not able to train the network beyond trivial performance. **(Right)** Jacobian Norm at initialization/convergence for the different scenarios reported on the left figure. The Jacobian norm was computed for a randomly picked example from the training set. Note that BN has no effect on the initial Jacobian norm since BN parameters are initialized as $(\beta_1, \beta_2) = (0, 1)$.

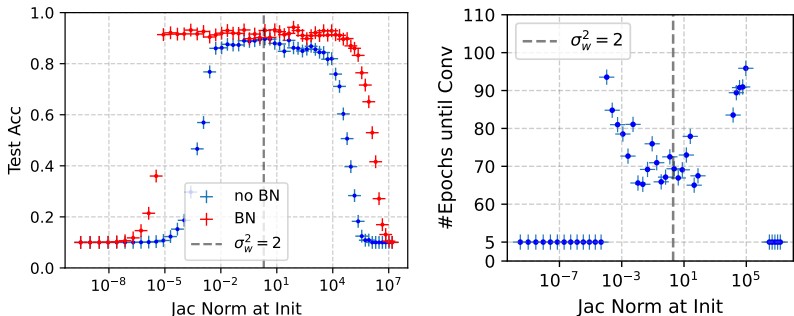

Figure 17: **(Left)** Test Acc after convergence Vs Jacobian norm at init for a VGG network of depth $L = 35$ trained with the same setup as in Fig 1 above. We vary the initialization hyperparameter $\sigma_w^2$ between $1$ and $3$ to obtain the range of values for the Jacobian norm. Large values of the Jacobian norm lead to numerical instability which makes the network untrainable with this setup. The effect of BN is noticeable in this case as it allows networks with exploding/vanishing Jacobian norms to be trainable. However, BN is not sufficient when the Jacobian norm is extremely large/small (typically of order 1e7/1e-7). **(Right)** Number of training epochs until convergence. Reasonable values of the Jacobian norm are associated with fast convergence (with non-trivial performance). When the network is untrainable, the number is fixed to $5$ (see the caption of Fig1 for definition of convergence).

