# OpenReview forum: "A Theoretical Study of the Jacobian Matrix in Deep Neural Networks"
_ICLR.cc/2024/Conference — ICLR 2024 Conference Withdrawn Submission_

### Official Review · Reviewer_dN56 · 2023-10-19

**Soundness:** 1 poor
**Presentation:** 3 good
**Contribution:** 1 poor
**Rating:** 3
**Confidence:** 4

**Summary:**

This paper studies the input-output Jacobian matrix during training. Assuming independence of weights across layers and infinite-width limit, it shows that the Jacobian matrix does not exponentially explode or vanish if it is initialized under the “stability of chaos” scheme. It further argues that if step size is sufficiently small, the Jacobian matrix norm remains nearly constant up to a certain time $t$. On a simple linear model, if the step size is of practical size, the Jacobian matrix norm was shown to monotonically decrease.

**Strengths:**

The behavior of the Jabobian matrix during training is an interesting and important question to study.

The paper is clearly written and well-structured.

This paper has a detailed discussion about existing results on the Jacobian matrix at initialization.

**Weaknesses:**

1. It is not clear whether Approximation 1 is an existing result that has been proved or an assumption. It is presented like an assumption, but in section 4 (also slightly mentioned in the abstract), it is claimed as a recent breakthrough from Random Matrix Theory. I hope the authors can clarify: what is exactly the “recent breakthrough”? how does Approximation 1 relate to this result.

2. This is one of my major concerns. The paper assumed (in Approximation 2) that the weights matrices are i.i.d. across layers. I don’t think this is true. Think about the simple case – a two layer linear network: $f= v^T u x$, where $u$ and $v$ are the first and second layer weights, and $x$ is the input. During training, $u$ and $v$ are clearly correlated, as $v^T u$ must change its value. This should also be true with non-linear activation. The authors probably mistakenly think that: the learn rate is of small order resulting in a small change in each weight, hence keeping independent weight matrices. However, that is not enough for independence.

3. In the discussion after Theorem 3, the paper claims that the required step size is “mild” and is generally satisfied in practice. I could not agree on this point. The theorem is in the infinite-width limit, i.e., $n\to\infty$, hence one can not ignore the factor $log(n)$, which should be very big resulting in an infinitesimal step size. To me, Theorem 3 is a trivial result, as the step size is so tiny that the training does not really go far away from the initialization.

4. The paper claims “the stagnation phase spans the whole training procedure” without justification. As I mentioned above, the so-called stagnation phase is probably due to the small learning rate so that the training has not gone far yet. It is totally unclear whether the stagnation phase lasts until convergence.

5. The descent phase is only shown on a very simple model – one-dimensional linear model. It seems the analysis does not extend to more complicated models, hence, it is unclear whether this phase happen for other models, e.g., neural networks.

Moreover, this descent phenomenon on linear models has been shown in a prior work [1]. Hence, I don’t see much novelty on this point.


[1] Aitor Lewkowycz, Yasaman Bahri, Ethan Dyer, Jascha Sohl-Dickstein, and Guy Gur-Ari. “The large learning rate phase of deep learning: the catapult mechanism”.

**Questions:**

no further questions

---

### Official Review · Reviewer_CMuq · 2023-10-24

**Soundness:** 3 good
**Presentation:** 2 fair
**Contribution:** 2 fair
**Rating:** 5
**Confidence:** 3

**Summary:**

This paper analyze the Jacobian matrix of deep neural network. Compared with previous works that analyze Jacobian stability at initialization, this paper studies Jacobian behaviour during training using recent tools of random matrix theory. The authors show that under some assumptions, the stability at initialization guarantees the stability during training. The authors also identify three phases of Jacobian behavior, including stagnation phase, decent phase and convergence phase. The authors also study the Jacobian in the setup of network pruning.

**Strengths:**

This paper focuses on analyzing the Jacobian behavior of deep neural networks, which is of great importance for both understanding DNN as well as for many real world applications. The authors develop a general framework to tackle this problem, using the recent results in random matrix theory. The authors conduct experiments to validate their theory.

**Weaknesses:**

1. The assumptions are strong and unrealistic. In Assumption 2, the authors assume that up to time $t$, the entries of the $D$ matrix follow i.i.d Bernoulli distribution, matrix $D$ are independent from the weight matrix $W$, and the weight matrix are independent across layers. On theoretical side, I can not see why these assumptions hold for even a toy model. On the practical side, although the authors conduct experiments on synthetic datasets to validate this, I am not convinced that these assumptions hold for the general case. Therefore, I can not tell whether the claimed Jacobian analysis beyond initialization comes from the overassumptions.
2. The classifications of Jacobian behavior into three phases is not well validated. For the first phase, the authors only provide theoretical justifications, but contradict the experiment results in figure 2, since the plots do not show that the Jacobian norm can stays approximately constant in the early phase of training. For the second and the third phase, the authors mainly provide empirical evidences but without a satisfactory theoretical justifications. I encourage the authors to conduct more extensive experiments to validate the claimed three stage behavior.
3. The presentations are not clear. For example, the contents are Section 4 and Section 5 are cut apart and I cannot see their internal connection upon first reading. I would recommend moving the stability theorem to the main text, explain the theorem before using it for Section 4 and Section 5. Furthermore, there are many typos that can cause misunderstanding of the main results. For example in Theorem 3. Assumptions 1 be assumption 2, if I understand it correctly. Overall, I recommend the authors to polish the writing and make a clear presentation of the results.

**Questions:**

See the weaknesses part.

---

### Official Review · Reviewer_wXpB · 2023-10-30

**Soundness:** 3 good
**Presentation:** 3 good
**Contribution:** 2 fair
**Rating:** 5
**Confidence:** 4

**Summary:**

Increasing the depth of neural networks can lead to gradient issues unless the right initialization is chosen. Previous research identified stable initializations by analyzing the input-output Jacobian. While past studies focused on initialization using random matrix theory, they expanded on this, exploring Jacobian behavior during training. They informally prove that stability during initialization continues through training by using new results in random matrix theory.

**Strengths:**

The subject explored in this paper is of profound interest. It poses an essential and inherent inquiry regarding the maintenance of stability for the Jacobian matrix in neural networks from initialization throughout training. This is the inaugural study on this particular topic.

The manuscript is well written and has a clear structure.

The paper employs novel methodologies to examine the spectrum of the Jacobian matrix. The findings from Bandeira et al. (2021) are indeed aptly integrated into the analysis of neural networks and warrant recognition within the learning theory community.

**Weaknesses:**

Assumption 2 appears to be quite stringent, as it presumes certain conditions throughout the entire training process. While there is empirical support provided, it lacks a solid theoretical basis, and the given intuitive reasoning doesn't resonate with me. This assumption seems crucial for this study.

Regarding Theorem 3, I'm not entirely certain, but my interpretation is that it essentially says: "In the initial stages of training, we are in the kernel regime with a stagnant feature, leading to a static norm of the Jacobian." Is the challenge in proving this mainly technical? That's how I currently see it. Furthermore, the example in Theorem 4 seems overly simplistic. Perhaps analyzing it using multi-dimensional input in deep linear networks would be more illuminating? Lastly, the assertion that gradient-based training inherently pushes the network towards smaller gradients, causing the Jacobian norm to decrease, doesn't sit right with me. In the end, even with a zero gradient, the Jacobian norm remains non-zero.

**Questions:**

Why haven't you detailed the RMT tools you used in the main content? Although you mention using new tools in the abstract, there isn't sufficient explanation in the main body of the text.

---

### Official Review · Reviewer_wG2F · 2023-11-01

**Soundness:** 3 good
**Presentation:** 4 excellent
**Contribution:** 2 fair
**Rating:** 5
**Confidence:** 4

**Summary:**

This paper is centered around studying the Jacobian of MLPs during training and at initialization. Compared to other studies on stability and signal propagation, the paper provides results on the evolution of the Jacobian through time. The authors first provide general results on the evolution of the Jacobian under gradient descent (3 phases) and then study a very interesting setting: Jacobian at initialization after pruning. Theoretical results are complemented with empirical evaluations confirming the correctness of the claims.

**Strengths:**

I generally like this paper: it combines theory with experimental evidence, and it's well-organized, well-written, and interesting. What I like most about it is the part about pruning – I believe that could be very important for applications. Also, the first 3 pages provide a very nice literature overview. While I have some points below about potential improvements, which I think are necessary to address for acceptance.

**Weaknesses:**

1) Evolution of the Jacobian: to me, the results are not surprising. If gradient descent can optimize the network, then it is clear the Jacobian would not blow up. That said, similar results for the 3 phases can be found here: https://arxiv.org/pdf/2106.16225.pdf (Figure 8, Lemma 8). While the paper above is on the Hessian rank, arguably, the effects are similar. But anyway, I like the fact that the authors are being more precise and that they provide bounds.

2) Figure 2: the discussion around these results, I think, is misleading. I am not able to spot a stagnation phase in any curve. What do you mean? I think here the interesting thing is the gap between Jacobians for methods that reach the same accuracy. However, I think one can draw a link to sharpness and say that the higher learning rate converges to flatter regions (for the edge of stability), and therefore, the Hessian (related to the Jacobian) will be smaller. I do not think your results can completely capture this phenomenon, and I would definitely not link this to feature learning and muP.

3) pruning: all makes sense, and again, I think this is very interesting. I think what is missing though is a trainability test: can you train your pruned networks after the jacobian has been stabilized?

Hope you can address some of the points above so I can raise my score!

**Questions:**

above

---

### Official Review · Reviewer_y7ZT · 2023-11-03

**Soundness:** 3 good
**Presentation:** 3 good
**Contribution:** 2 fair
**Rating:** 5
**Confidence:** 3

**Summary:**

The authors study the behavior of the Jacobian for deep neural networks during training. In particular, it is shown for multilayer perceptrons that if the Jacobian is stable at the initialization, under some mild conditions remains stable during training. The theory is extended to sparse networks considering two pruning techniques.

**Strengths:**

- Understanding the training dynamics of deep networks is a particularly interesting problem.
- The technical part of the paper seems solid, but I have not verified the theoretical results.

**Weaknesses:**

- The paper is written fairly well, but I think that Sections 3 and 4 do not communicate well the intended information/contributions.
- The main result of the paper is that if the Jacobian is stable at initialization, then for small enough learning rates it remains almost constant during training, mainly focusing on the infinite-width regime where lazy-training typically applies. Instead, for the non-infinite widths the Jacobian seems to change. I think it is not clear what is the actual goal of the paper.

**Questions:**

Please clarify the contributions and the goals of the paper. Especially, the fact that the theory focuses mainly on the infinite-width regime, while the Jacobian seems to change in the non-infinite regime.